# Discordant Humoral and T-Cell Response to mRNA SARS-CoV-2 Vaccine and the Risk of Breakthrough Infections in Women with Breast Cancer, Receiving Cyclin-Dependent Kinase 4 and 6 Inhibitors

**DOI:** 10.3390/cancers15072000

**Published:** 2023-03-27

**Authors:** Cristina Saavedra, Alejandro Vallejo, Federico Longo, Juan José Serrano, María Fernández, María Gion, Elena López-Miranda, Noelia Martínez-Jáñez, Eva Guerra, Jesús Chamorro, Diana Rosero, Héctor Velasco, Adrián Martín, Alfredo Carrato, José Luis Casado, Alfonso Cortés

**Affiliations:** 1Medical Oncology Department, Ramón y Cajal University Hospital, IRYCIS, 28034 Madrid, Spain; 2Laboratory of Immunovirology, Department of Infectious Diseases, Ramón y Cajal University Hospital, IRYCIS, 28034 Madrid, Spain; 3CIBERONC, Medical Oncology Department, Ramón y Cajal University Hospital, IRYCIS, Alcalá University, 28034 Madrid, Spain; 4CiberInfect, Infectious Disease Department, Ramón y Cajal University Hospital, 28034 Madrid, Spain

**Keywords:** SARS-CoV-2 vaccination, CDK4/6 inhibitors, cellular response, humoral response

## Abstract

**Simple Summary:**

Patients with cancer are at increased risk of serious COVID-19, thus, vaccination is especially relevant for this population. However, lower responses are expected, secondary to immunosuppression. Several studies have been conducted to determine how cancer patients respond to vaccination, but these studies include heterogenous groups of patients who received different types of treatment and had a variety of comorbidities that may influence COVID-19 susceptibility. This study aimed to determine the vaccination response in a specific subgroup of breast cancer patients receiving treatment with CDK4/6 inhibitors. Our study shows that SARS-CoV-2 vaccines are safe and effective in terms of humoral response in these patients, achieving antibody titers even higher than healthy controls. Surprisingly, the patients show an impaired cellular immune response, which could entail shorter protection. This finding should be considered for vaccination strategies.

**Abstract:**

Few data are available about the immune response to mRNA SARS-CoV-2 vaccines in patients with breast cancer receiving cyclin-dependent kinase 4/6 inhibitors (CDK4/6i). We conducted a prospective, single-center study of patients with breast cancer treated with CDK4/6i who received mRNA-1273 vaccination, as well as a comparative group of healthcare workers. The primary endpoint was to compare the rate and magnitude of humoral and T-cell response after full vaccination. A better neutralizing antibody and anti-S IgG level was observed after vaccination in the subgroup of women receiving CDK4/6i, but a trend toward a reduced CD4 and CD8 T-cell response in the CDK4/6i group was not statistically significant. There were no differences in the rate of COVID-19 after vaccination (19% vs. 12%), but breakthrough infections were observed in those with lower levels of anti-S IgG and neutralizing antibodies after the first dose. A lower rate of CD4 T-cell response was also found in those individuals with breakthrough infections, although a non-significant and similar level of CD8 T-cell response was also observed, regardless of breakthrough infections. The rate of adverse events was higher in patients treated with CDK4/6i, without serious adverse events. In conclusion, there was a robust humoral response, but a blunted T-cell response to mRNA vaccine in women receiving CDK4/6i, suggesting a reduced trend of the adaptative immune response.

## 1. Introduction

The COVID-19 pandemic spread globally, leading to 761 million confirmed cases and more than 6.8 million deaths throughout the world [1]. The introduction of SARS-CoV-2 vaccines has provided an important relief. However, since cancer patients were not included in pivotal vaccine trials, concerns have been raised about the safety and efficacy in this population.

Currently, there are some prospective data in cancer patients, concluding that solid-tumor patients need at least two, or even three, doses to achieve a satisfactory humoral response, whereas hematological patients usually show worse results [2,3,4,5,6,7,8]. However, there are conflicting results about the longevity of antibody response to SARS-CoV-2 [9], and studies on vaccinated individuals have mainly focused on serological response and neutralizing antibodies [10].

Cyclin-dependent kinase 4/6 inhibitors (CDK4/6i), in combination with hormonal therapies, constitute the standard treatment in luminal metastatic breast cancer. Therefore an important number of patients are treated with these therapies [11,12]. Secondary leukopenia and neutropenia occurs in up to 80% of patients, being grade 3 (less than 1000 neutrophils per microliter) and 4 (less than 500 neutrophiles per microliter) in 50% and 10% of the patients, respectively [13]. Therefore, the immune suppression caused by these therapies could potentially affect vaccination response and efficacy. On the contrary, preclinical data suggest that CDK4/6i could boost immunity due to the involvement of the CDK complexes in the development and activation of the immune system [14,15].

An observational study was performed to analyze the efficacy, in terms of humoral response and CD4+ and CD8+ cell responses, of COVID vaccination in patients with breast cancer under treatment with CDK4/6i, in comparison to healthy controls. The safety and differential toxicity profile were also studied. Here we report the results of patients undergoing treatment with CDK4/6i, and compare these data with the results obtained in a matched cohort of healthcare workers (HCW).

## 2. Materials and Methods

A prospective study was designed in patients with solid malignancies. The study was performed in accordance with Good Clinical Practice guidelines and the World Medical Association Declaration of Helsinki [16]. The study was approved by the institutional review board of Ramón y Cajal University Hospital (IRB 412/21) and all patients provided written informed consent before any procedure was carried out.

Eligibility criteria included age above 18 years, indication of vaccination according to clinical practice, and diagnosis of breast cancer with active CDK4/6i and endocrine therapy treatment (defining “active” as the last administration of the therapy up to 4 weeks before inclusion in the study). Exclusion criteria were evidence of previous SARS-CoV-2 infection. Individual patient data, from a prospective study carried out in health-worker volunteers vaccinated according to clinical practice with the BNT162b2 vaccine, were used as a control group, and matched according to age and sex, in order to minimize bias.

After inclusion, breast cancer patients received two doses of mRNA-1273 vaccine intramuscularly, 28 days apart. The vaccination program for cancer patients was designed by the public healthcare system authorities, and it was not possible to control the timing of the vaccine administration within the anticancer therapy schedule.

The objectives of the study were to assess the humoral (total IgG and neutralizing antibody titers) and CD4+ and CD8+ T-cell responses, in terms of frequency and magnitude after vaccination in the cohort of patients with CDK4/6i treatment, in comparison with healthy controls. The safety of vaccination was also assessed. Breakthrough infections were determined during follow-up.

Data collection was performed through the electronic clinical record. The following variables were compiled: age, sex, stage of malignant disease, oncologic therapy, the timing of vaccine administration in relation to CDK4/6i cycle, other comorbidities (presence or absence of hypertension, mellitus diabetes, dyslipidemia, cardiomyopathy, pneumopathy, thrombosis), active smoking habit and adverse events (presence or absence of local pain, local erythema, local edema, fatigue, headache, myalgia, arthralgia, vomiting, diarrhea, chills, fever), COVID infection after vaccination and vital status. After the completion of vaccination, the patients were followed-up for 12 months.

Three blood samples were obtained from each patient: before the administration of the first vaccine dose (baseline, BL), at least 21 days after the administration of the first vaccine dose (1D) and three-to-four weeks after the second vaccine dose (2D). The study design allowed us to investigate the kinetics of immune responses following primary and secondary immunization. The complete study design is shown in Figure 1A.

For the different analysis, 30 mL of venous blood was obtained in EDTA tubes, and processed within 2 h after the collection. After centrifugation, the plasma fraction was stored at −80 °C, while the cellular fraction was diluted in phosphate-buffered saline (PBS), followed by a Ficoll density gradient centrifugation for the isolation of PBMCs, which were subsequently washed and frozen with fetal bovine serum (FBS) and 8% dimethyl sulfoxide (DMSO).

Antibodies against the SARS-CoV-2 nucleocapsid (COVID-19-SARS-CoV-2 IgG ELISA, Demeditech, Germany; positivity threshold 11 relative units (RU)/mL) were measured in the BL sample to identify those with a previous infection. Specific antibodies for the SARS-CoV-2 spike protein (SARS-CoV-2 IgG II Quant Alinity; Abbott, Maidenhead, UK; positivity threshold 50 arbitrary units (AU)/mL) were measured in all the samples.

Following the WHO recommendation for the standardization of anti-SARS-CoV-2 immunoglobulin determination, the antibody level units were converted to binding-antibody units (BAU)/mL (conversion factor 0.142) [17]. Consequently, for specific antibodies to SARS-CoV-2 spike protein, the positivity threshold was equivalent to 7.14 BAU/mL.

SARS-CoV-2-neutralizing antibodies were quantified using the competitive inhibition enzyme immunoassay technique (Human Novel Coronavirus (SARS-CoV-2) Neutralizing Antibody ELISA Kit, MyBioSource), following the manufacturer’s instructions. Plate wells were pre-coated with SARS-CoV-2 RBD, and horseradish peroxidase-conjugated ACE2 was added to the sample. The competitive inhibition reaction was launched between HRP–ACE2 and SARS-CoV-2-neutralizing antibodies in the samples. A substrate solution was added to the wells and the color developed in an inverse manner to the amount of SARS-CoV-2-neutralizing antibodies in the sample. Optical densities greater than half the optical density of the blank were considered negative. Results were recorded as ng/mL.

Cellular immune response was assessed in the baseline sample and after completion of the immunization regimen. Briefly, after the gating of singlet cells, lymphocytes were morphologically selected, and then CD3 T cells were gated. Cell debris, monocytes, and B cells were excluded from the analysis, and live CD3 T cells were selected. IFN-g expression was finally analyzed separately for CD4+ and CD8+ T cells. SARS-CoV-2-specific CD4+ and CD8+ T cells were measured using in vitro stimulation with SARS-CoV-2 peptide pools of viral protein spike (S), followed by quantitation of CD4+ and CD8+ T-cell-specific interferon (IFN)-γ by cell flow cytometry, using peripheral blood mononuclear cell (PBMC) samples from all subjects. The test was considered reactive if the proportion of positive cells in the stimulated wells was at least 2-fold higher in comparison with the negative control wells (unstimulated). Flow cytometry is illustrated in Appendix A.

Concerning the statistical analysis, continuous variables were expressed as the median and interquartile range (IQ25-75) and categorical variables by frequencies and proportions. As this is an exploratory study, statistical analysis to estimate the necessary number of patients to include, was not performed.

Comparisons between the groups were performed using two-tailed statistical tests, chi-square or Fisher’s exact tests, for categorical variables, and a Mann–Whitney test or one-way analysis of variance (Kruskal–Wallis test) with Dunn’s correction for multiple comparisons, as appropriate. Paired samples were compared using the Wilcoxon-signed rank test. Statistical significance was defined as two-sided *p* values < 0.05. Statistical analysis was performed with the STATA version 15 software, and GraphPadPrism version 8 for figures.

## 3. Results

An observational prospective study was performed on 35 women with breast cancer with an active CDK4/6i treatment in combination with endocrine therapy, who were scheduled to receive the mRNA-1273 SARS-CoV-2 vaccine. As mentioned earlier, patients with a previous history of COVID-19 were excluded.

Patients included in the study had a baseline determination of humoral and T-cell response that was repeated at least 28 days after the first (humoral response) and second dose (humoral and cellular responses). From the 35 patients initially included in the CDK4/6i cohort, nine patients had a positive baseline serology and were excluded from further analysis. The second blood sample was not obtained in one patient from the CDK4/6i group due to a deterioration in her performance status (see final study flow diagram in Figure 1A).

At the same time, a cohort of 26 HCWs, matched by age and sex, were selected as controls, also after excluding those with a COVID-19 history or positive basal serology. Patient characteristics are shown in Table 1.

In the CDK4/6i group, the three approved drugs were represented, although the number of patients taking each treatment was unbalanced (nine patients received abemaciclib, 15 palbociclib, and two ribociclib). Most of the patients received the first dose of the vaccine during treatment (25 patients).

Regarding humoral response, all patients reached the threshold value of positivity. However, CDK4/6i patients achieved higher levels of anti-S IgG antibodies after the first vaccine dose (*p* = 0.045), although non-significant differences were found after the second vaccine dose compared to HCWs (Figure 1B). Neutralizing antibody titers were similar between CDKi patients and HCWs (*p* = 0.161, Figure 1C).

Neutralizing antibody titers had a positive correlation with anti-S IgG antibodies across all samples (R^2^ = 0.258, *p* < 0.001), as shown in Figure 1D. 

Nearly one third of the patients in each group had a T-cell response at baseline, suggesting the presence of cross-reactivity in the absence of prior disease or positive serology. Of note, the rate and magnitude of T-cell response at baseline were similar in both groups (Figure 2A,B). Strikingly, the rate of individuals with spike-specific CD4+ and CD8+ T-cell response after vaccination was 19% lower in the CDK4/6i group versus the HCW group, although these differences were not statistically significant. Furthermore, the magnitude of both CD4+ and CD8+ T-cell responses in the CDK4/6i group was significantly lower after two doses of the mRNA vaccination (Figure 2A,B).

During follow-up, the percentage of infection was slightly higher in the CDK4/6i group compared to HCW (19% vs. 12%, *p* = 0.703). Although viral-variant sequencing was not possible, the main variant during follow-up was the alpha V1, with an increment of the Omicron variable at the end of this period. Albeit non-significant, there was a lower magnitude of T-cell response in those with breakthrough infections during follow-up. Notably, lower levels after the first dose of vaccine were detected in those individuals who had COVID-19 after vaccination (492.29 vs. 157.96 BAUs/mL; *p=* 0.029), with a tendency of lower titers of neutralizing antibodies (*p* = 0.068).

A level of anti-S IgG antibodies after the first dose of vaccine below 366.77 BAUs predicted a higher risk of subsequent SARS-CoV-2 infection, with a sensitivity of 71.43% and a specificity of 72.73% (area under curve, AUC 0.766, 95% CI 0.615–0.892). Furthermore, IgG levels after the second dose were associated with breakthrough infections (AUC 0.614, 95% CI, 0.470–0.747). A similar association was observed with the titers of neutralizing antibodies after the first vaccine dose (sensitivity 75%, specificity 70.45%; AUC 0.705, IC 0.569–0.829). On the other hand, no significant correlations were found between CD4 and CD8 T-cell responses and neutralizing or IgG antibody titers (Figure 3A,B).

Although non-significant, a lower level of CD4 T-cell response was found in individuals with post-vaccination breakthrough infections, and a similar level of CD8 T-cell response, regardless of breakthrough infections (Figure 4A). While no differences were found in the IgG antibody titers according to breakthrough infections, a surprisingly higher level of neutralizing antibody titers were found in CDKi patients with breakthrough infections (Figure 4B).

Finally, the overall incidence of adverse events (AEs) was higher in the group of patients under treatment with CDK4/6i. Local symptoms, headache and chills were the most frequents AEs in the CDK4/6i group versus the HCW group. No serious adverse events were reported in either of the two groups (Table 2 and Figure 5).

## 4. Discussion

Immunosuppression secondary to cancer and oncologic therapies increases the risk of severe COVID-19 and, at the same time, decreases the effect of the vaccines [2,3,4,5,6,7]. However, cancer patients are a heterogeneous group of patients with different neoplasms, grades of comorbidities and treatments. Thus, specific data about the real impact of each disease and each oncologic therapy on vaccine efficacy are still needed.

A prospective study was performed to analyze the humoral and cellular response to vaccination in breast cancer patients receiving treatment with CDK4/6i and endocrine treatment, in comparison with a group of healthy volunteers. We observed that patients with CDK4/6i had a better humoral response in comparison to the HCW group. Thus, anti-S IgG titers and neutralizing antibodies were numerically higher in the CDK4/6i group and there was a significant correlation between both serological parameters [18].

This fact contrasts with the previous assumption of a lower response to vaccination in cancer patients. In a similar study with 21 breast cancer patients receiving CDK4/6i, Zagouri et al. [19] showed a similar neutralizing antibodies response. Mechanisms underlying neutropenia with CDK4/6i are different than those with chemotherapy [20]. In fact, despite the high rates of grade 3–4 neutropenia with CDK4/6i, the rate of infections and febrile neutropenia are extremely low, because leukopenia and neutropenia are induced through cell-cycle arrest of bone marrow precursors [13,21]. Therefore, it is not surprising to find a better humoral immune response to the vaccine in CDK4/6i patients in comparison with the response found in other patients with cancer receiving other treatments [2,4,22].

For the first time in this group of patients, we show that the CDK4/6i cohort had a lower T-cell response compared with HCWs, even considering the possible role of cross-reactivity at baseline. The main limitation in interpreting these results is the lack of agreement about the T-cell threshold value that is considered protective [6,23]. In addition, the mechanisms responsible for the 19% lower response rate and significantly lower magnitude are unclear.

Although leukopenia and neutropenia are well-known adverse events of these inhibitors, previous studies indicate that CDK4/6i could have a relevant role in promoting immune checkpoint-inhibitor action. It is well known that cyclins and CDKs control the development, differentiation and activation of immune cells [14,15]. Paradoxically, in vitro analysis has demonstrated the promotion of a memory versus an effector signature in CD8 T cells exposed to CDK4/6i, which is independent of cell-cycle arrest, since it is not reproducible with other cell-cycle inhibitors, and these results were replicated in vivo in a small cohort of breast cancer patients and healthy donors [24,25] Thus, our unexpected results highlight the importance of more studies concerning T-cell response after vaccination. Besides, the percentages of lymphocytes (obtained from PBMCs), and CD4 and CD8 T cells (obtained from live CD3+ lymphocytes) were similar in both groups at baseline and after vaccination.

As previously reported, patients who developed breakthrough infections during follow-up tended to have lower anti-S IgG and neutralizing antibody titers [18,26]. In recent studies in patients with hematological malignancies, obtaining anti-S IgG ≥ 300 BAUs/mL was considered adequate, as it represents the lower level of antibodies obtained in healthy individuals, and correlates with potent virus neutralization [27]. A similar threshold was used in patients with cancer [28]. Only one patient (4%) in our cohort had an initial value below the cutoff of 300 BAUs (94 BAU/mL), suggesting the waning of the humoral response during follow-up to explain incident infections. We also showed an association between lower T-cell magnitude and incident infections during follow up, suggesting the failure of an adaptative immune response and confirming the need for the combination of the humoral and cellular response.

Finally, there was a higher incidence of adverse events in the CDK4/6i group. Symptoms related with the disease or oncologic treatment could influence the rate of adverse events in this population, but most of them were local reactions and also systemic symptoms, such as fatigue, headache, myalgias or chills.

Our study has several limitations. The most important is related to the modest number of patients and controls included, which limits the generalizability of the results. Second, the lack of another control group of patients with breast cancer who were not taking CDK4/6i, is a limitation. Third, the study was limited to the analysis after the second vaccine dose, and not extended to the third or fourth vaccine dose to ascertain the role of new infections with different variants of concern. Finally, we included patients with cross-reactive cellular immunity at baseline that could have influenced the final rate of T-cell responses, although they were similar in both cohorts at baseline [29,30].

## 5. Conclusions

In conclusion, patients treated with CDK4/6i showed a robust humoral response, but a blunted trend of T-cell response in comparison with a cohort of healthy volunteers, suggesting a trend of reduced adaptative immune response in this group. Moreover, a higher risk of infection was observed during follow-up in those patients with lower humoral response after vaccination. Our data support the need for additional studies in this subgroup of patients to establish vaccination strategies.

## Figures and Tables

**Figure 1 cancers-15-02000-f001:**
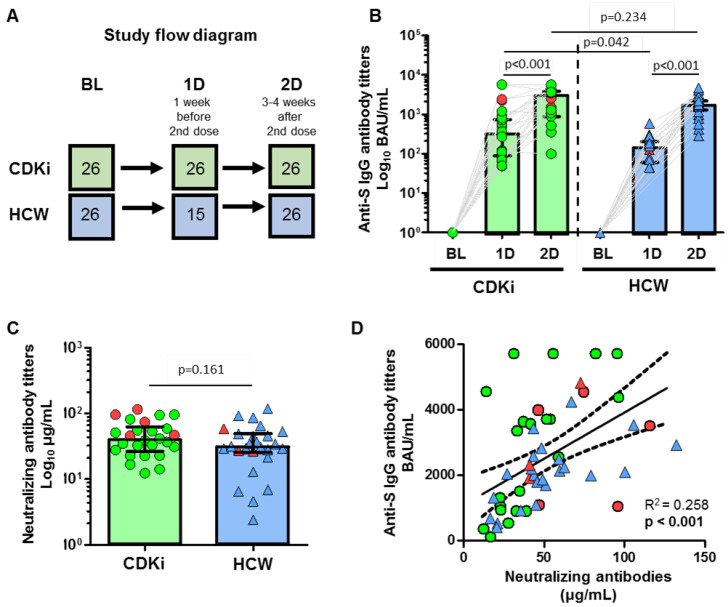
(**A**) Study flow diagram showing both cohorts: breast cancer patients receiving CDK4/6 inhibitors (CDKi) and healthcare workers (HCW). The sample extraction points were at baseline (BL), after the first vaccine dose (1D), and after the second vaccine dose (2D). The number of patients at each point is also shown. (**B**) Comparison of anti-SARS-CoV-2 spike IgG antibody titers between CDKi patients (green circles) and HCWs (blue triangles) at the three time points. (**C**) Comparison of neutralizing antibody titers between CDKi patients (green cicles) and HCWs (blue triangles) after the second vaccine dose. (**D**) Correlation between anti-SARS-CoV-2 spike IgG and neutralizing antibody titers in CDKi patients (green circles) and HCWs (blue triangles). Individuals among both cohorts who had post-vaccination breakthrough infections are highlighted in red. A Spearman’s test was used for statistical analysis. Data were considered significant when *p* < 0.05.

**Figure 2 cancers-15-02000-f002:**
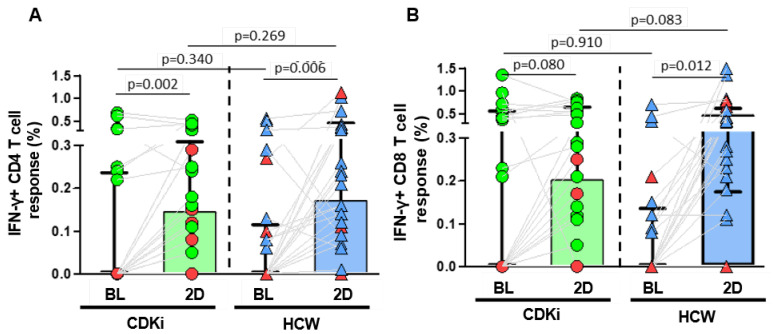
SARS-CoV-2 spike-specific T-cell response in breast cancer patients receiving CDK inhibitors (CDKi, green circles) and healthcare workers (HCW, blue triangles). (**A**) CD4 T-cell response at baseline and after vaccination in CDKi and HCWs. (**B**) CD8 T-cell response at baseline and after vaccination in CDKi and HCWs. Individuals among both cohorts who had post-vaccination breakthrough infections are highlighted in red.

**Figure 3 cancers-15-02000-f003:**
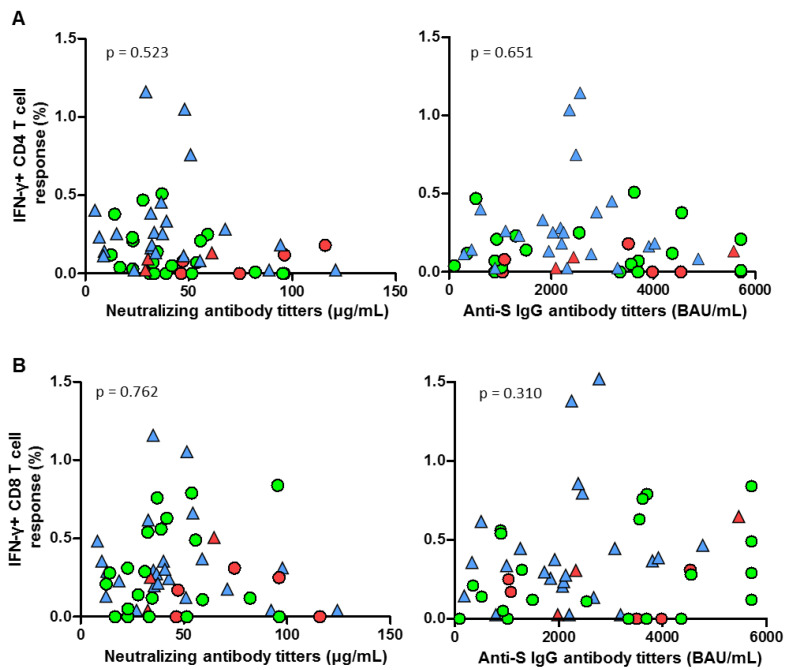
(**A**) Correlation between spike-specific CD4 T-cell response and both anti-S IgG (left) and neutralizing (right) antibody titers in both breast cancer patients receiving CDK inhibitors (CDKi, green circles) and healthcare workers (HCW, blue triangles). (**B**) Correlation between spike-specific CD8 T-cell response and anti-S IgG (left) and neutralizing (right) antibody titers in both CDKi patients (green circles) and HCWs (blue triangles). Individuals among both cohorts who had post-vaccination breakthrough infections are highlighted in red. A Spearman’s test was carried for statistical analysis. Data were considered significant when *p* < 0.05.

**Figure 4 cancers-15-02000-f004:**
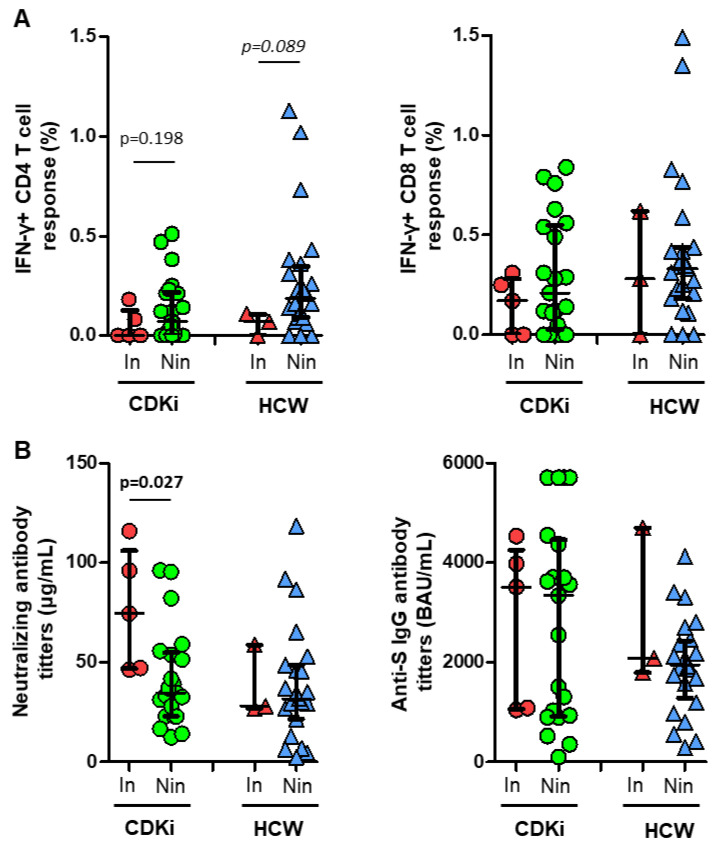
(**A**) Comparison between CD4 (left) and CD8 (right) T-cell responses, according to post-vaccine breakthrough infections (In, infected in red; and Nin, not infected in green or blue) in both breast cancer patients receiving CDK inhibitors (CDKi, green circles) and healthcare workers (HCW, blue triangles). (**B**) Comparison between neutralizing (left) and anti-S IgG (right) antibody titers, according to post-vaccine breakthrough infections (In, infected in red; and Nin, not infected in green or blue) in both CDKi patients (green circles) and HCWs (blue triangles). A Mann–Whitney U test was used for statistical analysis, and data were considered significant when *p* < 0.05.

**Figure 5 cancers-15-02000-f005:**
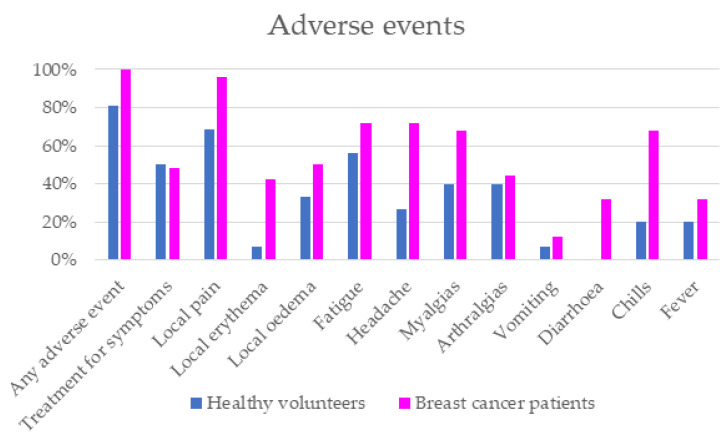
Incidence of adverse events.

**Table 1 cancers-15-02000-t001:** Patient characteristics.

		CDK4/6 Inhibitors (*n* = 26)	Healthy Control Workers (*n* = 26)	*p*-Value
Age		53.3 (29.1–77)	48.6 (33.6–65.9)	0.19
Female		100%	100%	1
Stage IV		96%	-	
Comorbidities				
	Hypertension	27%	8%	0.14
	Diabetes	7.70%	4%	1
	Dyslipidemia	15%	24%	0.499
	Heart chronic disease	7.70%	4%	1
	Pulmonary chronic disease	0%	8%	0.235
	Smoking habit	31%	52%	0.16
Anti-S IgG titers (BAU/mL) at BL		0.85 (0–2.6)	4.94 (3.79–6.01)	0.001
Humoral immunogenicity (anti-S IgG, BAU/mL)				
	1D Positive individuals	366 (111–806) 100%	164 (70–236) 100%	0.042 1
	2D Positive individuals	3427 (1026–4372) 100%	1946 (1573–2455) 100%	0.2 1
Neutralizing antibodies 2D (IgG, µg/mL)		40 (27–59)	31 (26–48)	0.158
Lymphocytes (%) at BL		66.9 (50.9–77.2)	63.4 (58.5–68.7)	0.742
	CD4 T cells (%)	63.7 (56.4–69.4)	64.3 (59.1–70.3)	0.756
	CD8 T cells (%)	28.9 (20.7–34.3)	23.1 (18.8–31.1)	0.122
T-cell cross-reactivity at BL				
	Individuals with Anti-S CD4 T cell response	23.10%	30.80%	0.755
	Individuals with Anti-S CD8 T cell response	34.60%	34.60%	1
Lymphocytes (%) after 2D		74.1 (64.4–78.2)	67.9 (61.1–73.6)	0.129
	CD4 T cells (%)	67.4 (62.7–73.9)	66.7 (63.7–70.2)	0.552
	CD8 T cells (%)	27.2 (21.3–32.1)	28.1 (24.7–31.6)	0.647
T-cell response after 2D				
	Anti-S CD4 (%)	0.12 (0–0.25)	0.16 (0.09–0.31)	0.269
	Individuals with CD4 response	69%	85%	0.324
	Anti-S CD8 (%)	0.155 (0–0.31)	0.32 (0.18–0.44)	0.083
	Individuals with CD8 response	69%	85%	0.324

Data are expressed as the median (interquartile range) and proportion of positive cases when indicated. BL = baseline. 1D = after first vaccine dose. 2D = after second vaccine dose. % lymphocytes of peripheral mononuclear cells (PBMCs); % CD4/CD8 T cells of CD3+lymphocytes.

**Table 2 cancers-15-02000-t002:** Incidence of adverse events.

	CDK4/6 Inhibitors (*n* = 26)	Healthy Control Workers (*n* = 26)	*p*-Value
Any adverse event	100%	81.25%	0.049
Treatment for symptoms	48%	50%	>0.05
Local pain	96.15%	68.75%	0.023
Local erythema	42.31%	6.67%	0.03
Local oedema	50%	33.33%	>0.05
Fatigue	72%	56.25%	>0.05
Headache	72%	26.67%	0.009
Myalgias	68%	40%	>0.05
Arthralgias	44%	40%	>0.05
Vomiting	12%	6.67%	>0.05
Diarrhea	32%	-	-
Chills	68%	20%	0.008
Fever	32%	20%	>0.05

## Data Availability

The data generated and analyzed during this study are described in the data record. The immune response and clinical data are available in the Excel spreadsheet “BD_ONCO_300621.xlsx”, which is openly available and shared as part of the figshare data record. The “ONCOVAC study” dataset is not publicly available to protect patient privacy. Requests for access to this dataset can be made to the corresponding author.

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
