# Peer review of "Discordant Humoral and T-Cell Response to mRNA SARS-CoV-2 Vaccine and the Risk of Breakthrough Infections in Women with Breast Cancer, Receiving Cyclin-Dependent Kinase 4 and 6 Inhibitors"

_cancers, 2023, doi:10.3390/cancers15072000_

Round 1
Reviewer 2 Report
Dear Authors, I am pleased that your work tries to address immune responses to the COVID-19 mRNA vaccine in cancer patients receiving CDK4/6 inhibitor therapy. This is very timely work given that these patients may be at high risk of immunosuppression and subsequent high risk of infection including COVID-19, especially in patients whose disease progressed on therapy.
However, the data presented herein is exploratory in nature and thus lacks details. Your conclusion might have been supported by more convincing and relevant findings generated by a well-designed study which includes:
1. Positive control: breast cancer patients on non-CDK4/6 inhibitor treatment. This is because it has already been established by recent preclinical and clinical studies that CDK4/6 inhibitors boost the patient's own immunity to fight cancer. They do so by activating the effector T cells, improving antigen presentation, and suppressing the immune suppressors (such as Tregs, and myeloid-derived suppressor cells). Your study lacks all these details and thus needs to be provided before this work is considered for publication in the nominated journal.
2. RNA seq data showing the immune profiles of the patients vs normal control to know the major pathways that have undergone a change in relation to adaptive immune systems to capitalize on your claims and conclusion.
Your background should discuss the role of CDK4/6 in immunomodulation and how their inhibition could be harnessed in the COVID-19 mRNA vaccine situation.
Reviewer 3 Report
In their study, Saavedra and al. are reporting the important, but also surprising finding that breast cancer patients on CDK4/6i therapy respond to an mRNA-based COVID vaccine as well, if not better, than a reference cohort. Studies such as this one are an important contribution to the body of literature that provides support for the notion that cancer patients can, after all, greatly benefit from these vaccines, although these results go far beyond COVID vaccines.
Overall, this is a straight-forward, entirely observational study. The manuscript would greatly benefit from being edited by a native/fluent English speaker for grammar and to correct confusing and awkward statements (a few of these issues are listed under “Grammatical or English-language issues”), since they make it hard to read the manuscript.
The main concern is related to the major finding of this study – the (albeit not significant) increase in antibody titers and neutralizing activity, despite the (striking) decrease in – especially - CD4 T cell activity. While this was not designed to be a mechanistic, but an exploratory observational study, the authors should still address these seemingly contradictive results (eg in the discussion and/or the limitations-section) to help direct follow-on studies: an impairment of the CD4 T (helper) cell compartment would be expected to negatively impact the humoral response, resulting in issues such as decreased antibody titers, decreased avidity (which could affect neutralizing activity), altered Ig-subclass distribution, or shortened memory responses. The fact that the authors observed somewhat better humoral immunity in the patients could have different reasons, for example that the “wrong” compartment (peripheral T cells) was sampled and that tissue resident Tfh were actually increased in numbers and/or that CD4 T cells were sequestered in lymphatic tissues. Alternatively, the reported neutropenia could have had an impact on (relative) T cell numbers in the periphery and the authors should make it clear what the total T cell numbers were in the donors (Fig. 1E/F – how was the percentage of responding T cell calculated?) It appears that calculations were based on unstimulated controls only, but T cell numbers were not normalized.
Since the authors emphasize the finding here and in other studies that total (and neutralizing) antibody titers against SARS-CoV-2 are more likely to be associated with breakthrough infections, while the role (and threshold) of T cells is unclear, it would be helpful to include an additional figure in which antibody titers are correlated with (especially) CD4 T cell responses (and, as recommended already for Fig 1D, highlighting the individuals with subsequent breakthrough infections). Clarifying which individuals experienced breakthrough infections is particularly relevant for this study since the impact of the strength of the vaccine-induced response is part of the manuscript’s title.
The issues listed below are mainly minor and the authors should be able to easily address them.
- Line 21: “Several studied have been conducted [that] focused on….”: the implications of the different treatments and comorbidities should at least be mentioned briefly (ie, suppression of immune responses and higher susceptibility to infection) in this summary. Also, the statement “As we expected…. “ (line 23) should be avoided because it implies study bias.
- Line 25: replace “they” with “the patients”, and “This” (line 26) with something like “This finding”.
- The statement in Line 56 could be confusing: the authors indicate that a combination of CDK4/6i with hormone therapy is the standard of care for breast cancer therapy, but the description of their patient cohort only mentions CDK4/6i. Thus, it would be helpful to clarify whether (and if no, why) their patients did not receive hormone therapy since this might have an impact on vaccine-induced immune responses.
- Line 85: should be “total IgG and neutralizing antibody titers” instead of “neutralizing antibodies and IgG antibodies”
- Line 89 and/or Line 96: should specify the duration of the follow up (during which data on breakthrough infections were collected) to gauge the potential impact of the treatment on the durability of the vaccine-induced immune response (and whether breakthrough infections occurred before individuals had received their subsequent booster vaccinations). Also, COVID vaccine-efficacy is impacted by the viral variant that the vaccinee is exposed to. Assuming that the authors did not sequence the viral isolates from breakthrough infections, it would still be important to indicate what the dominant variants were (in the area where the study was conducted) at the time the study was conducted.
- Line 124: “opposite” is the wrong choice of words here!
- Line 127: unclear what “at inclusion” means! Also, should be “after completion of the immunization regimen”
- Line 131: what is “live cell flow cytometry”? For intracellular staining, the cells would have had to be fixed so they aren’t alive.
- Line 148: should be “were scheduled to receive”, or “who had planned to receive” (instead of “who were planned to receive”)
- Line 153: “baseline” instead of “basal” (same in line 157)
- Line 153: unclear what “performance status worsening” means
- Fig 1D: at first glance, the results do not seem to match those shown in panels B and C – in those earlier panels, there are no low-response “outliers” when it comes to total IgG responses, and also very tight neutralizing antibody titers in the patient cohort, however, in panel D, the responses appear to be much more heterogenous (the use of different scales certainly contributes to the confusion). The authors need to clarify whether the data in both, panels C and D are from after the first or second immunization (since panel B shows all three time points). Also, considering the statement in the manuscript that lower antibody titers were associated with breakthrough infections, it would be helpful if the individual data points were color coded (ie, green/blue to indicate which cohort the data point is associated with, as well as with an additional color – eg red outline – for those with breakthrough infections).
- Line 191: need to clarify “lower levels” of what?
- Line 195: shouldn’t this read “a level of anti-S IgG….. BELOW 366 BAUs”?
- Line 198: since, presumably, not all individuals with IgG levels below the threshold experienced breakthrough infections, this should be “correlated” instead of “were associated”. Again, this finding is surprising considering that the patient cohort (according to Fig1C) had more consistent neutralizing antibody titers compared to the HCW-cohort, which also had, in total, slightly lower titers. This should be addressed/discussed.
- Line 210: the authors refer to a variety of studies that have reported negative impacts of cancer therapy on vaccine efficacy, however, do not cite any specific studies here. Should add at least a few relevant references.
- Line 219: unclear what “both components of humoral response” means – should this be “both serological parameters measured in this study”?
- Line 268: the statement “associated with a lower response after vaccine” (should be “vaccination”) is rather vague and the authors should specify whether this only refers to the serological parameters measured or also to T cell responses,
Grammatical or English-language issues:
- Line 20: “Several studies have been conducted focused” should be something like “Several (or many) studies have been conducted to determine how cancer patients respond to vaccination, but these studies tend to include heterogenous groups of patients who received different types of treatment and had a variety of comorbidities”.
- Line 32: Should be “Higher neutralizing antibody titers and anti-……were observed…”
- Line 47: COVID vaccines “provided” relief (instead of “led to relief)
- Line 48: should be “concerns have been raised about the safety and efficacy of COVID vaccines in this population” (instead of “concerning about….”)
- Line 53: should be “responses to SARS-CoV-2”, instead of “response in SARS-CoV-2”
- Line 58: “exposed” is not a suitable term here!
- Line 91: it is unclear what “time in the cycle of vaccine administration” means – please rephrase!
- Line 102: should be “analyses”, not “analysis”
- Line 108: should be “At BL, we measured antibodies against the SARS-CoV-2 NP….”
- Line 111: the phrasing of this sentence is confusing and suggests that no anti-S protein antibody measurements were done at BL. Instead of saying “in the three CONSECUTIVE samples”, authors should say something like “in all samples”. Also, should be “were measured”, not “was measured”
- Line 244: “Accordingly with previously demonstrated” needs to be rephrased! Eg., “As previously reported”
- Line 247: “lower level antibodies” needs to be replaced with something like “lower antibody titers”
- Line 249: unclear what the sentence “A similar level….” means. Needs to be rephrased.
Round 2
Reviewer 1 Report
Most of the issues have been fixed/clarified.
I have two final suggestions:
1-As the included cancer patients were not previously vaccinated, then previous vaccination should be mentioned among the exclusion criteria just like previous infection.
2- Add to the limitations section the issue of using two different vaccines in the 2 groups (control and patients) and clarify the reason as you explained in your response to the comments.
Author Response
"Please see the attachment."

Reviewer 2 Report
Dear Authors,
Thank you for getting back with your feedback. It is understandable that you might not able to get controls and RNA seq facilities. However, you could still have performed detailed immune profiling by flow cytometry only to support your conclusion with stronger evidence. Still, the appropriate account of CDK4/6 in immunomodulation is lacking in your introduction.
Author Response
"Please see the attachment."

Reviewer 3 Report
The authors were highly responsive to feedback and critiques and have addressed all the concerns raised. I only have the following, very minor follow-up comments:
- Line 47: the term “emergence” should be reserved for pathogens, not products like vaccines (recommend something like “introduction” or “rollout”). Also, instead of “provide”, should be “has provided”
- Line 65: should be “responses” (plural)
- Line 147: should be “was performed on 35 women” (not “of women”)
- Line 174: should be “antibody titers” not “antibodies titers”
- Line 174: please clarify that the R and p value are for all samples (
- Line 175: assuming that the authors mean “and the close correlation between the two parameters is shown in figure 1D”?
- Line 184: should be “titers”, not “titters” (same typing error in line 216, 221, 227, 236), and “Individuals from both cohorts…”
- Please be consistent how “T cells” is spelled (should be without the hyphen, but the manuscript uses both versions)
- Figure 2: The (new) figure legend indicates that “individuals from both cohorts who had breakthrough infections are highlighted in red”, however, none there are no red symbol in Fig 2!
- Recommend re-structuring sentence in line 283: those with breakthrough infections tended to have lower antibody titers (since not ALL patients with lower antibody titers also had breakthrough infections)
- Line 282: should be “after vaccination” (instead of “after vaccine”)
- Line 286: similar to my previous comment - the term “represents the lower level antibodies” needs to be fixed (and clarified)
- Line 175: Ig titers and neutralizing activity are “aspects of the humoral response”, not “components of the humoral responses”
Author Response
"Please see the attachment."

Round 3
Reviewer 2 Report
The topic is highly interesting. However, your conclusions and claims haven't yet been supported by your findings. There should be appropriate controls for your study otherwise this causes an ethical concern.